# Differentiate Soybean Response to Off-Target Dicamba Damage Based on UAV Imagery and Machine Learning

**Caio Canella Vieira** [1,†], **Shagor Sarkar** [2,†], **Fengkai Tian** [2], **Jing Zhou** [3], **Diego Jarquin** [4], **Henry T. Nguyen** [2], **Jianfeng Zhou** [2] and **Pengyin Chen** [1,*]

1   Fisher Delta Research, Extension, and Education Center, Division of Plant Science & Technology, University of Missouri, Portageville, MO 63873, USA; cc2k8@umsystem.edu
2   Division of Plant Science & Technology, University of Missouri, Columbia, MO 65211, USA; ssw5r@umsystem.edu (S.S.); ft7b6@umsystem.edu (F.T.); nguyenhenry@missouri.edu (H.T.N.); zhoujianf@umsystem.edu (J.Z.)
3   Biological Systems Engineering, University of Wisconsin-Madison, Madison, WI 53706, USA; jzmc3@umsystem.edu
4   Agronomy Department, University of Florida, Gainesville, FL 32611, USA; jhernandezjarqui@ufl.edu
*   Correspondence: chenpe@missouri.edu
†   These authors contributed equally to this work.

**Abstract:** The wide adoption of dicamba-tolerant (DT) soybean has led to numerous cases of off-target dicamba damage to non-DT soybean and dicot crops. This study aimed to develop a method to differentiate soybean response to dicamba using unmanned-aerial-vehicle-based imagery and machine learning models. Soybean lines were visually classified into three classes of injury, i.e., tolerant, moderate, and susceptible to off-target dicamba. A quadcopter with a built-in RGB camera was used to collect images of field plots at a height of 20 m above ground level. Seven image features were extracted for each plot, including canopy coverage, contrast, entropy, green leaf index, hue, saturation, and triangular greenness index. Classification models based on artificial neural network (ANN) and random forest (RF) algorithms were developed to differentiate the three classes of response to dicamba. Significant differences for each feature were observed among classes and no significant differences across fields were observed. The ANN and RF models were able to precisely distinguish tolerant and susceptible lines with an overall accuracy of 0.74 and 0.75, respectively. The imagery-based classification model can be implemented in a breeding program to effectively differentiate phenotypic dicamba response and identify soybean lines with tolerance to off-target dicamba damage.

**Keywords:** soybean; dicamba; RGB; UAV; machine learning

## 1. Introduction

Soybean (*Glycine max* (L.) Merr.) represents the largest and most concentrated segment of global agricultural trade [1]. The growing demand for soybean is primarily attributed to its unique seed composition and versatile applications in the food, feed, and biodiesel industries as the crop delivers the highest amount of protein per hectare and accounts for over 60% of total global oilseed production [2,3]. Worldwide, soybean production has reached over 384 million metric tons, of which Brazil (144 million metric tons, 37.5%), the United States (120 million metric tons, 31.3%), and Argentina (49.5 million metric tons, 14.4%) account for roughly 85% of the global production [4].

Proper weed management is essential to sustain soybean production, with significant yield reductions as high as 53% being observed when fields are left untreated [5–7]. Integrated weed management systems rely on a combination of mechanical, cultural, chemical, and biological practices to minimize environmental impact and potential development

of herbicide-resistant weed populations [8]. With the development and commercialization of genetically engineered soybean cultivars resistant to over-the-top applications of dicamba (3,6-dichloro-2-methoxybenzoic acid, DT) [9,10], approximately 55% of the soybean acreage in the United States have quickly adopted the technology [11]. This rapid and widespread adoption has resulted in numerous reports of off-target dicamba damage to non-DT soybean fields as well as multiple dicots plant species [12–16]. Soybean is naturally susceptible to dicamba, and the consequential symptoms include crinkling and cupping of the immature leaves, epinasty, plant height reduction, chlorosis, death of apical meristem, malformed pods, and, ultimately, yield reduction [17–19]. The severity of the observed symptoms and yield penalty vary depending on the growth stage, dosage, frequency, and duration of exposure, and potentially genetic background, of which soybean is two to six times more susceptible to dicamba when exposed at the early reproductive stage [19–25].

The assessment of injuries caused by off-target dicamba exposure is generally reported as categorical variables (tolerant, moderate, susceptible) or percentage of injury (0–100%) based on visual observations. Such assessment is time-consuming, labor-intensive, and is often subjective to the evaluator, which can result in biased and inconsistent ratings [26,27]. Plant breeders often investigate tens of thousands of breeding lines for specific or multiple phenotypes in a growing season [3]. The development of an accurate and high-throughput platform to characterize breeding materials is highly desired. Remote sensing technology is a cost-effective approach to identify and quantify changes in plant biophysical and biochemical properties that has been widely applied in agricultural research [28,29]. These biophysical and biochemical changes in plants can be identified by UAV-image-derived features such as vegetation indices (VIs) and plant geometric features [30]. Vegetation indices can be generated from RGB [31], multispectral [29,32], and hyperspectral [33] images to identify plant vigor and vegetation coverage under different stress conditions [34]. In soybean, multiple studies targeting assessment and quantification of herbicide injuries using image-derived features have been reported, including injuries caused by dicamba [33,35–37], 2,4-D (2,4-Dichlorophenoxyacetic acid) [36], glyphosate [38–40], and metribuzin (4-amino-6-*tert*-butyl-3-(methylthio)-1,2,4-triazin-5(4*H*)-one) [41]. However, all previous studies have been based on a limited number of soybean cultivars with narrow genetic diversity. Therefore, the goals of this research were to (i) develop an RGB image-based classification system using machine learning algorithms and seven image features, including canopy coverage, contrast, entropy, green leaf index (GLI) [42], hue, saturation, and triangular greenness index (TGI) [43], with a total of 230 diverse soybean breeding lines and 10 commercial cultivars (seven DT and three glyphosate-tolerant (GT)) and (ii) assess and compare classification accuracies between artificial neural network (ANN) and random forest (RF) algorithms. The development of a simple and cost-effective RGB-based classification system may allow plant breeders to precisely and rapidly screen and effectively select genotypes tolerant to off-target dicamba damage.

## 2. Materials and Methods

### 2.1. Plant Materials and Data Acquisition

A total of 230 diverse soybean breeding lines developed by the University of Missouri Fisher Delta Research, Extension, and Education Center (MU-FDRC) soybean breeding program and 10 commercial cultivars (seven DT and three GT) were used in this study. Breeding lines were derived from 115 unique biparental populations and ranged from relative maturity 4.0 to 5.3. These lines comprised the 2020 advanced yield trials at the MU-FDRC, and a subgroup of lines selected from the 2019 advanced yield trials based on extreme response to off-target dicamba damage (tolerant and susceptible) and yield performance under prolonged off-target dicamba exposure.

Field trials were conducted at the Lee Farm in Portageville, MO (36°23′44.2″N; 89°36′52.3″W) using a randomized complete block design with three replications per environment. The Lee Farm has been exposed to season-long off-target dicamba damage since 2017, where non-DT breeding lines often experience significant yield losses compared

to the DT commercial checks [44–46]. The 2020 advanced yield trials were grown in three environments (Fld-61, Fld-63, Fld-81) and the subgroup of extreme lines was grown in two locations (Fld-86, Fld-1210) (Table 1). Each plot consisted of four rows 3.66 m long, spaced 0.76 m apart.

**Table 1.** Field trials conducted to develop a UAV-based classification model for off-target dicamba response.

| Location | Trial [1] | #Entries [2] | #Plots [3] | Planting | Imaging | DAP [4] | Visual Scoring | DAP |
|---|---|---|---|---|---|---|---|---|
| Fld-61 | AYT | 213 | 670 | 04/17/2020 | 08/20/2020 | 125 | 8/20/2020 | 125 |
| Fld-63 | AYT | 213 | 670 | 04/28/2020 | 09/08/2020 | 133 | 9/9/2020 | 134 |
| Fld-81 | AYT | 213 | 672 | 04/18/2020 | 08/21/2020 | 125 | 8/21/2020 | 125 |
| Fld-86 | Subset | 48 | 144 | 06/01/2020 | 09/15/2020 | 106 | 9/14/2020 | 105 |
| Fld-1210 | Subset | 48 | 144 | 05/27/2020 | 09/14/2020 | 110 | 9/14/2020 | 110 |

[1] Trial: "AYT" corresponds to the 2020 advanced yield trials and "Subset" corresponds to the selected group of soybean lines from the 2019 advanced yield trials based on extreme (tolerant and susceptible) response to off-target dicamba. [2] #Entries: Number of unique soybean lines visually and digitally phenotyped for off-target dicamba damage. The total number of entries among trials (261) exceeds 230 due to the overlapping of soybean lines between trials. [3] #Plots: Total number of plots visually and digitally phenotyped for off-target dicamba damage. Plot number does not necessarily equal unique entries x replications, due to replicated genotypes and/or deactivated plots. [4] DAP: Days after planting, number of days for data collection after planting.

### 2.2. Visual Dicamba Damage Assessment

Field plots were visually assessed for the dicamba damage at early reproductive stages between R3 to R5 depending on the line's maturity group (approximately 100 to 130 DAP) [47]. Lines were rated as tolerant, moderate, and susceptible based on the severity of dicamba symptoms (Figure 1). The tolerant group showed similar plant development to the DT commercial checks with none to minimal visual dicamba damage, including the typical crinkling and cupping of the immature leaves, reduced canopy area, and plant stunting. The moderate group showed intermediate dicamba damage symptoms, including mild crinkling and cupping of the immature leaves with minimal reduction in canopy area and plant height. The susceptible group exhibited extreme dicamba damage symptoms, including severe crinkling and cupping of the immature leaves and severe reduction in canopy area and plant height.

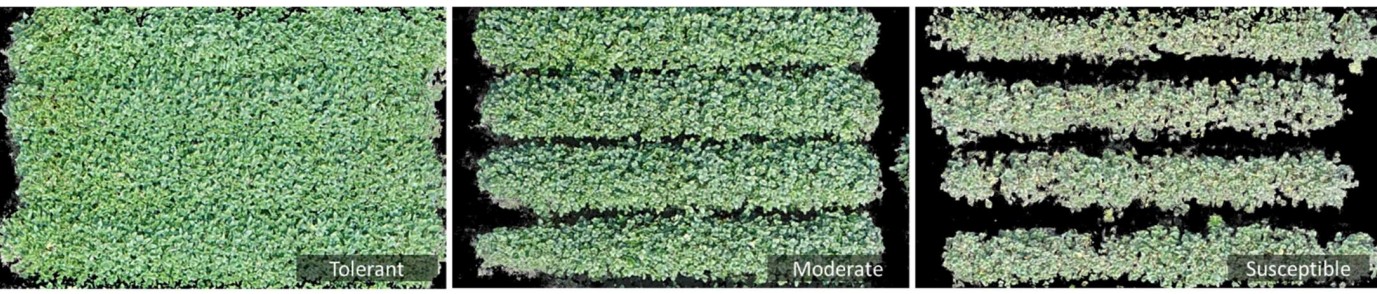

**Figure 1.** Ground-based classification scale of field plots to off-target dicamba damage with three differential phenotypic classes including tolerant, moderate, and susceptible. In this figure, the tolerant plot is the conventional breeding line S16-12774C, moderate is the conventional breeding line PR17-482, and susceptible is the GT commercial check AG 4135 (Monsanto Co., Creve Coeur, MO, USA).

### 2.3. UAV Imagery Data Acquisition

Field plot images were collected using a DJI Phantom 4 Pro (Version 1.0, DJI, Shenzhen, China) quadcopter. The quadcopter has a built-in RGB camera mounted onto a gimbal underneath the copter. The camera has an image resolution (number of total pixels) of 5472 × 3648 pixels and was configured to take time-lapse images at 2 frames per second (fps). The embedded global navigation satellite system (GNSS) receiver provides geo-referencing information as a part of metadata to each image frame. The UAV platform

was controlled using the flight control mobile app Autopilot (Hangar Technology, Austin, TX, USA) to complete flight missions autonomously by following predefined flight plans. Images were taken at 20 m above ground level (AGL) at the speed of 7 km/h, following a zigzag path to cover each field with the forward overlap of ≥70% and side overlap of ≥65%. The ground sampling distance (GSD) of the camera in this setting was 5.5 mm/pixel. UAV images were collected at noon in each field under a clear sky.

### 2.4. Image Processing and Features Extraction

An orthomosaic image for each field was generated using Agisoft MetaShape Pro (Agisoft LLC, St. Petersburg, Russia) following the methodology described by Zhou et al. (2019) [48]. Three parameters were set as "high" with generic and reference preselection for image alignment, "high" for reconstruction parameter, and "moderate" for filtering mode [48]. The orthomosaic for each field was generated and exported as .tif images and processed using the Image Processing Toolbox and Computer Vision System Toolbox of MATLAB (ver. 2020a, The MathWorks, Natick, MA, USA).

Individual field plots were separated from the orthomosaic image by manually cropping a rectangle region of interest (ROI) around each plot. The ROI size varied to cover each soybean plot according to its width and length. Overlapping between adjacent plots was avoided based on visual quality control. Background (soil, shadow, and plant residues) was removed from the separated images by detecting projected canopy contours using the "*activecontour*" function [49] with the "Chan–Vese" method [50]. Pixels within a full contour were considered as foreground (soybean plants), while those outside contours were background (soil and residues). Contours with extremely small regions were identified as noises using the "*regionprops*" function and then removed from the foreground.

Seven image features were calculated from the processed RGB images, including canopy coverage, color (hue, saturation (Sa)) in HSV color space, image texture (entropy and contrast), and two vegetation indices including *TGI* [43] and *GLI* [42] as defined in Equations (1) and (2).

$$TGI = \frac{(\lambda_{Red} - \lambda_{Blue})(\rho_{Red} - \rho_{Green}) - (\lambda_{Red} - \lambda_{Green})(\rho_{Red} - \rho_{Blue})}{2} \quad (1)$$

$$GLI = \frac{(Green - Red) + (Green - Blue)}{(2 \times Green) + Red + Blue} \quad (2)$$

where lambda ($\lambda$) = center wavelengths for the respective bands including red (670 nm), blue (480 nm), and green (550 nm); rho ($\rho$) = pixel value for the respective bands including red (670 nm), blue (480 nm), and green (550 nm).

Canopy coverage was defined as the total number of pixels in the green channel of each RGB image. The hue and saturation were calculated from the HSV color space converted from the RGB images using the function "*rgb2hsv*" in MATLAB. Following the protocol described by Zhou et al. (2020) [32], the image texture entropy and contrast were calculated using the "*graycomatrix*" function in MATLAB after converting each RGB image to a grayscale level by the function "*rgb2gray*". Entropy typically quantifies the level of randomness and complexity of an image and can be used to characterize the texture of the image, of which larger entropy indicates higher complexity [51]. Since field plots image collection and visual assessment of off-target dicamba damage were conducted across different fields under variable environmental lighting conditions, UAV-image-derived features were standardized before inclusion in the ANN and RF predictive models. A z-score for each image feature was used for standardizing the model's predictors by dividing the difference between the observed value and the mean by the standard deviation (Equation (3)).

$$Z - score = \frac{x - \mu}{S} \quad (3)$$

where $x$ = observed value for an image feature in an individual plot; $\mu$ = mean of all the plots in an individual field for a specific image feature; $S$ = standard deviation of all the plots in an individual field for a specific image feature.

*2.5. Feature Significance*

A two-way analysis of variance (ANOVA) with an honest significant difference (HSD) Tukey test was conducted to investigate the significance of the difference between the image features and visual dicamba damage assessment among the five fields (Fld-61, Fld-63, Fld-81, Fld-86, Fld-1210). The two-way ANOVA test was generated with a 5% significance level by using the "*aov*" function original from R [52]. The Tukey's range test was performed using the "*TukeyHSD*" function from the *agricolae* package [53].

*2.6. Classification Algorithms and Accuracy*

In this study, an ANN model was used to classify soybean responses to off-target dicamba based on image features and ground visual damage assessment. The model was built using the "*neuralnet*" function of the R package "*neuralnet*" package [54] with five hidden layers, 1,000,000 iterations, and an error tolerance of 0.02, along with all other settings set to default. Additionally, an RF model was also used to classify soybean response to off-target dicamba based on image features. The model was built using the "*randomForest*" function of the R package *randomForest* [55] with the parameters "ntree" = 400, "mtry" = 2, and all other settings set to default. The RF model was configured to output the variable importance during the training process.

Performance of the ANN and RF models were assessed using a five-fold cross-validation method, which is a conventional model's accuracy evaluation and is commonly used in cases with a limited number of observations [56]. The three classes of visual dicamba damage scores (tolerant, moderate, and susceptible) were evaluated based on the number of samples correctly classified as true positive (*TP*), falsely classified as false positive (*FP*), correctly not classified as true negative (*TN*), and falsely not classified as false negative (*FN*) for both ANN and RF models. The overall accuracy was calculated using Equation (4). Class accuracy, which represents the ratio of correctly predicted instances and all the instances, was calculated using Equation (5). Precision, which indicates the proportion of predicted presences, was calculated using Equation (6), and sensitivity and specificity, which indicate the ratio of correctly predicted positive and negative classes, respectively, were calculated using Equations (7) and (8).

$$Overall\ Accuracy = \frac{No.\ of\ samples\ classified\ correctly\ in\ a\ test\ set}{Total\ No.\ of\ samples\ in\ a\ test\ set} \times 100\% \quad (4)$$

$$Class\ Accuracy = \frac{TP + TN}{TP + TN + FP + FN} \quad (5)$$

$$Precision = \frac{TP}{TP + FP} \quad (6)$$

$$Sensitivity = \frac{TP}{TP + FN} \quad (7)$$

$$Specificity = \frac{TN}{TN + FP} \quad (8)$$

where *TP* = true positive; *TN* = true negative; *FP* = false positive; *FN* = false negative.

**3. Results**

*3.1. Distribution of Visual Dicamba Damage Scores*

Across 2300 field plots, approximately 26.7% (614) were visually classified as tolerant, 36.4% (837) as moderate, and 36.9% (849) as susceptible (Figure 2). The overall distribution of scores was balanced. Unbalanced distributions were observed in locations Fld-86 and Fld-1210, where the most rated class was tolerant. This could be attributed to reduced off-target

dicamba exposure, late planting dates resulting in favorable environmental conditions to support plant recovery, and/or experimental error associated with the subjective visual assessment of the damage.

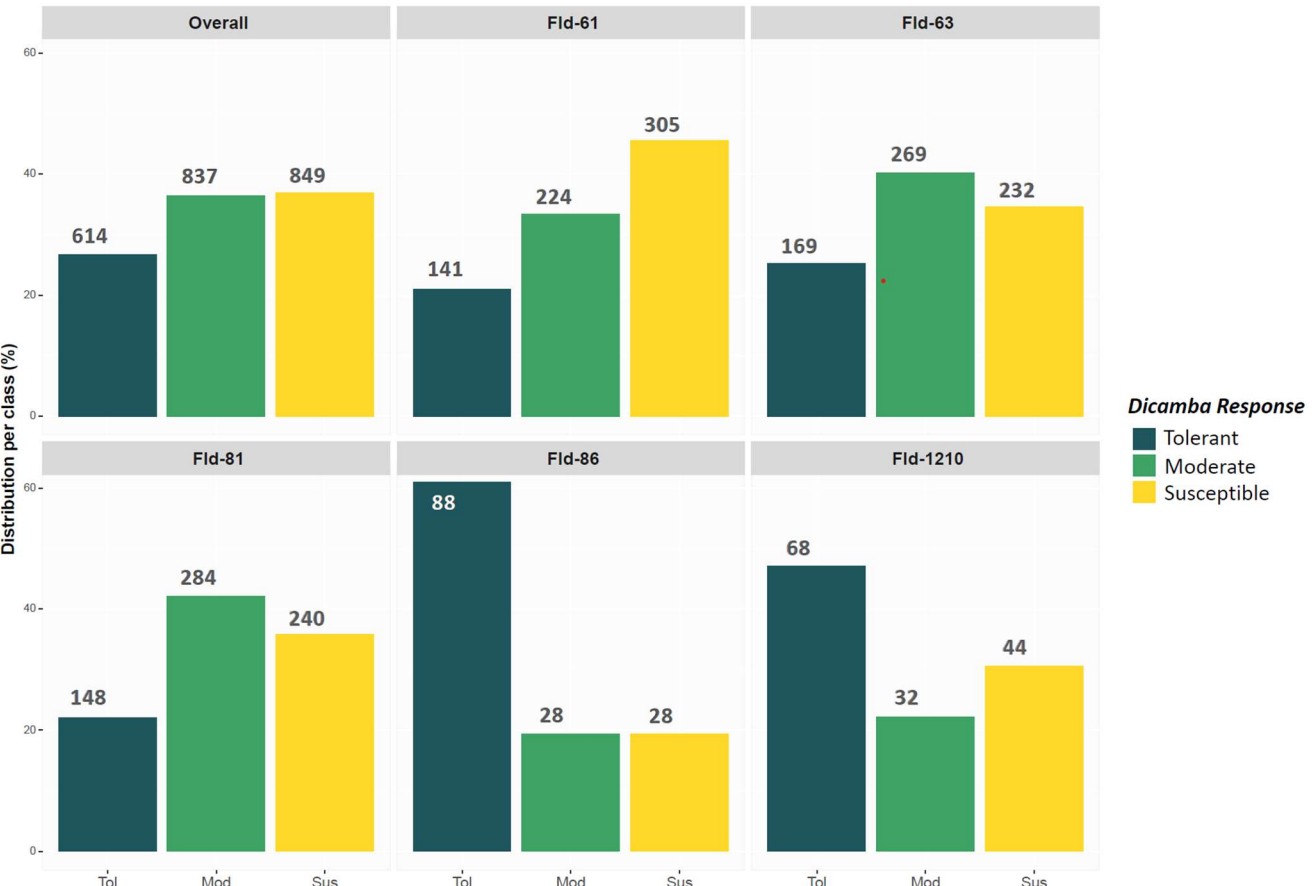

**Figure 2.** Distribution of dicamba response classification in each location and combined across all locations. Overall, approximately 26.7% of the plots were classified as tolerant, 36.4% as moderate, and 36.9% as susceptible.

### 3.2. Image Features across Classes of Visual Dicamba Damage Scores

A significant distinction among classes of dicamba response was observed across the seven image features (Table 2). Canopy coverage, entropy, GLI, Sa, and TGI significantly differentiated the three classes, whereas contrast and hue significantly differentiated tolerant and moderate classes from susceptible but did not have significant differences among tolerant and moderate classes. Although the tested fields represented diverse and unique environments (soil types and physical locations), no significant differences among fields were detected across all image features, which reinforces the consistency and importance of these features in differentiating tolerant, moderate, and susceptible soybean lines.

Overall, the observed values of each image feature across dicamba response classes were aligned well with the field observations and expected distributions (Figure 3). Higher values for canopy coverage, entropy, GLI, hue, Sa, and TGI indicate tolerance to dicamba, whereas higher values of contrast indicate susceptibility (Figure 3). Higher values of canopy coverage are expected for the tolerant class primarily due to healthy vegetative growth and minimal to no cupping of the immature leaves. Entropy, as a measurement of image complexity, represents the texture of each plot. Tolerant soybean, primarily due to the lack of cupping of the immature leaves, are logically perceived as homogenous and smoother and therefore have higher values of entropy. In contrast, susceptible soybean, as a consequence of intense cupping of the immature leaves due to dicamba damage, would

have an uneven and rough appearance and therefore show lower values of entropy. The GLI and TGI are vegetation indices that represent overall plant health and therefore are expected to be higher in the tolerant group and lower in the susceptible group. Saturation and hue represent the color structure of the image and indicate the intensity of the observed color. The cupping of the leaves affects the overall color reflection of the plant under sunlight, of which lighter tones of low-intensity green become predominant. Therefore, lower values of hue and Sa are observed in susceptible soybean as compared to the tolerant and moderate classes.

**Table 2.** Summary and significance of seven image features across dicamba response classes.

| Image Feature | Tolerant [1] | | Moderate | | Susceptible | | Field [2] |
|---|---|---|---|---|---|---|---|
| Canopy Coverage | 0.616 | a | 0.191 | b | −0.709 | c | N.S |
| Contrast | −0.151 | b | −0.191 | b | 0.531 | a | N.S |
| Entropy | 0.512 | a | 0.267 | b | −0.856 | c | N.S |
| GLI | 0.455 | a | 0.252 | b | −0.798 | c | N.S |
| Hue | 0.211 | a | 0.232 | a | −0.654 | b | N.S |
| Sa | 0.295 | a | 0.044 | b | −0.222 | c | N.S |
| TGI | 0.472 | a | 0.204 | b | −0.686 | c | N.S |

[1] Grouped letters represent significant distinction among dicamba response classes obtained through Tukey's HSD test (0.05). [2] N.S., non-significant, indicates that the variable "Field" was not significant and therefore the observed value for each feature in each response class across all locations was not significantly different.

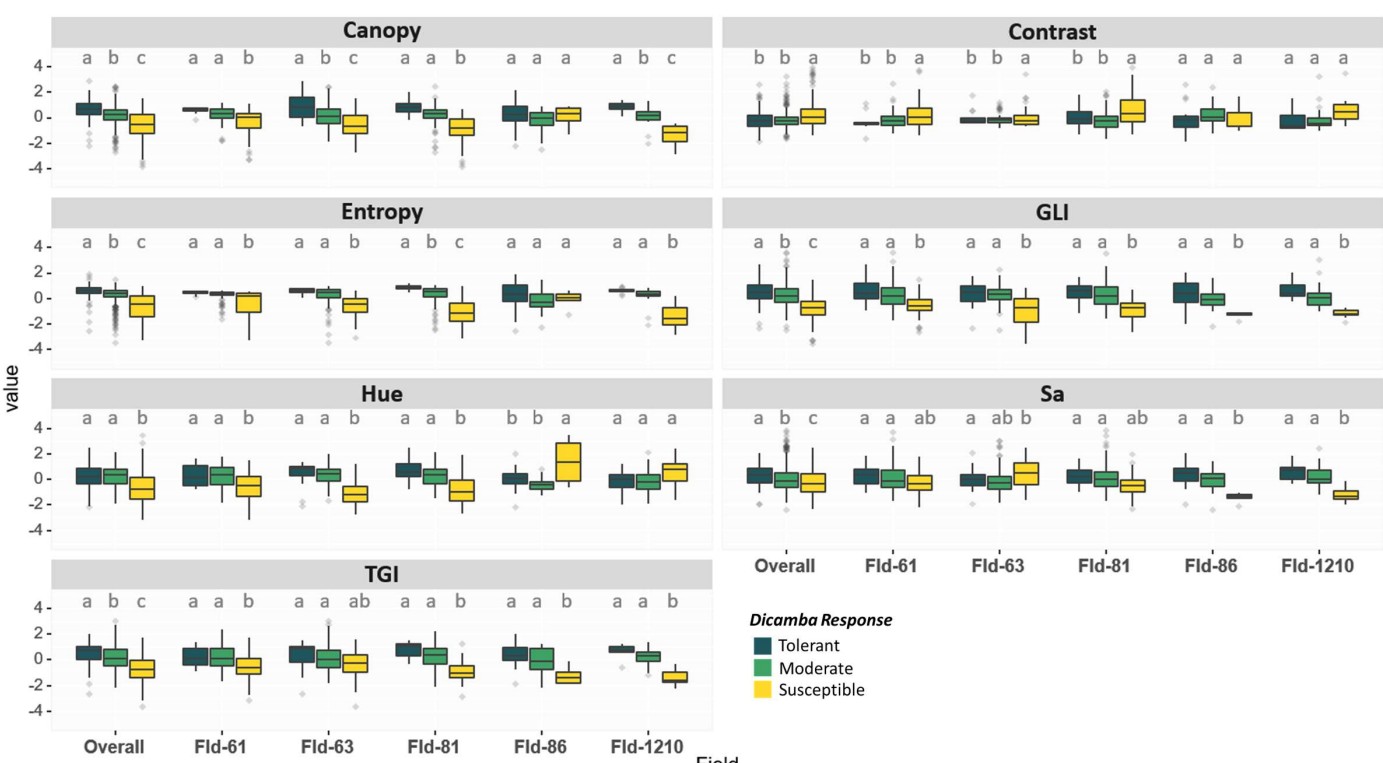

**Figure 3.** Standardized features distribution across all fields and dicamba response classes. Overall, significant differences among classes were identified for all features, of which higher values for canopy coverage, entropy, GLI, hue, Sa, and TGI indicate tolerance to dicamba, whereas higher values of contrast indicate susceptibility.

### 3.3. Model Performance and Overall Classification Accuracy

#### 3.3.1. Artificial Neural Network Model Classification

An artificial neural network model was used to classify the different classes of dicamba scores, and the model, based on interactive machine learning using all seven image features



as predictors, showed an overall classification accuracy of 0.74 with a five-fold cross-validation, whereas the highest accuracy was observed in Fld-63 (0.78) and lowest in Fld-61 (0.69) (Table 3). The locations Fld-86 and Fld-1210 were not included in the individual analysis due to the reduced sample size but were included in the overall analysis. The purpose of the classification in each field was to investigate whether a limited number of samples in individual fields can precisely classify different responses to dicamba. The model showed high accuracy for the classes tolerant (0.89) and susceptible (0.84) and slightly lower accuracy for the moderate class (0.75). Similarly, high specificity was observed for both tolerant and susceptible classes (0.97 and 0.91, respectively), and low specificity for the moderate class (0.54). Although these metrics showed promising high performance of the ANN model, the precision for the tolerant class (0.41) was substantially lower than both moderate (0.77) and susceptible (0.71) classes. Interestingly, the ANN model often misclassified true-moderate soybean as tolerant but rarely misclassified true-tolerant as susceptible (and vice versa). This indicates that, although the ANN model struggles to differentiate between moderate and tolerant, hence the low precision values, it can clearly distinguish the two extremist classes (tolerant and susceptible). Practically, this distinction is most important for soybean breeding and is adequate in helping breeders to make selections.

**Table 3.** Confusion matrix and model's performance metrics for dicamba response classification using RGB-based image features and ANN classifier.

| Dicamba Class | Overall [1] | | | Fld-61 | | | Fld-63 | | | Fld-81 | | |
|---|---|---|---|---|---|---|---|---|---|---|---|---|
| | Tol | Mod | Sus | Tol | Mod | Sus | Tol | Mod | Sus | Tol | Mod | Sus |
| Tolerant | 12 | 15 | 2 | 0 | 5 | 0 | 5 | 7 | 2 | 9 | 9 | 0 |
| Moderate | 53 | 410 | 68 | 17 | 122 | 26 | 20 | 140 | 10 | 19 | 133 | 13 |
| Susceptible | 7 | 46 | 120 | 2 | 16 | 25 | 0 | 7 | 22 | 0 | 9 | 21 |
| Class Accuracy | 0.89 | 0.75 | 0.84 | 0.89 | 0.70 | 0.79 | 0.86 | 0.79 | 0.91 | 0.87 | 0.77 | 0.90 |
| Precision | 0.41 | 0.77 | 0.71 | 0.00 | 0.74 | 0.58 | 0.36 | 0.82 | 0.76 | 0.50 | 0.81 | 0.70 |
| Sensitivity | 0.17 | 0.87 | 0.63 | 0.00 | 0.85 | 0.49 | 0.20 | 0.91 | 0.65 | 0.32 | 0.88 | 0.62 |
| Specificity | 0.97 | 0.54 | 0.91 | 0.97 | 0.39 | 0.89 | 0.95 | 0.49 | 0.96 | 0.95 | 0.48 | 0.95 |
| Overall Accuracy [2] | 0.74 | | | 0.69 | | | 0.78 | | | 0.77 | | |

[1] Overall is the combined analysis including Fld-61, Fld-63, Fld-81, Fld-86, and Fld-1210. [2] Overall accuracy is the average of five-fold cross-validation results.

### 3.3.2. Random Forest Model Classification

The overall classification accuracy using the RF model and a five-fold cross-validation was 0.75, whereas the highest accuracy was observed in Field-63 (0.80) and lowest in Fld-61 (0.71) (Table 4). Similar to the ANN model, locations Fld-86 and Fld-1210 were not included in the individual analysis due to reduced sample size but were included in the overall analysis. Classes tolerant and susceptible showed high accuracy (0.89 and 0.84, respectively) but, as opposed to the ANN model, the moderate class had slightly higher accuracy (0.77 and 0.75, respectively). Interestingly, the RF model was able to better distinguish the response classes, particularly tolerant and susceptible soybean. The precision for the tolerant class increased by 51% (0.62 vs 0.41) and 5% for the susceptible class (0.74 vs 0.71) compared to the ANN model. Overall, the model classified only three true-susceptible as tolerant (1.8%) and none of the true-tolerant lines were classified as susceptible. Similar to the ANN model, the RF can precisely distinguish between tolerant and susceptible soybean but with a slight advantage in precisely classifying the tolerant class.

**Table 4.** Confusion matrix and model's performance metrics for dicamba response classification using RGB-based image features and RF classifier.

| Dicamba Class | Overall [1] | | | Fld-61 | | | Fld-63 | | | Fld-81 | | |
|---|---|---|---|---|---|---|---|---|---|---|---|---|
| | Tol | Mod | Sus | Tol | Mod | Sus | Tol | Mod | Sus | Tol | Mod | Sus |
| Tolerant | 18 | 11 | 0 | 0 | 1 | 0 | 4 | 0 | 0 | 0 | 1 | 0 |
| Moderate | 51 | 420 | 69 | 9 | 114 | 34 | 8 | 137 | 21 | 14 | 129 | 17 |
| Susceptible | 3 | 40 | 121 | 0 | 18 | 37 | 2 | 11 | 30 | 0 | 16 | 36 |
| Class Accuracy | 0.89 | 0.77 | 0.85 | 0.95 | 0.71 | 0.76 | 0.95 | 0.81 | 0.84 | 0.93 | 0.78 | 0.85 |
| Precision | 0.62 | 0.78 | 0.74 | 0.00 | 0.86 | 0.52 | 0.29 | 0.93 | 0.59 | 0.00 | 0.88 | 0.68 |
| Sensitivity | 0.25 | 0.89 | 0.64 | 0.00 | 0.73 | 0.67 | 1.00 | 0.83 | 0.70 | 0.00 | 0.81 | 0.69 |
| Specificity | 0.98 | 0.54 | 0.92 | 0.96 | 0.66 | 0.78 | 0.95 | 0.77 | 0.88 | 0.93 | 0.69 | 0.89 |
| Overall Accuracy [2] | 0.75 | | | 0.71 | | | 0.80 | | | 0.77 | | |

[1] Overall is the combined analysis including Fld-61, Fld-63, Fld-81, Fld-86, and Fld-1210. [2] Overall accuracy is the average of five-fold cross-validation results.

### 3.3.3. Random Forest Feature Importance

The mean decrease Gini coefficient was calculated for the seven image features included in the RF model (Figure 4). The coefficients indicate how much accuracy the model loses by excluding each variable, of which the higher values of mean decrease Gini indicate higher importance of the variable in the model. The results show that hue, entropy, GLI, and TGI were the most important features in the classification model. Considering that the visual assessment of dicamba damage was primarily based on the overall appearance of field plots, including cupping of the immature leaves (texture represented as entropy), color intensity (represented as hue), and overall crop development and healthiness (represented as GLI and TGI), the results obtained from the decrease Gini coefficient align with field observations and feature importance expectations.

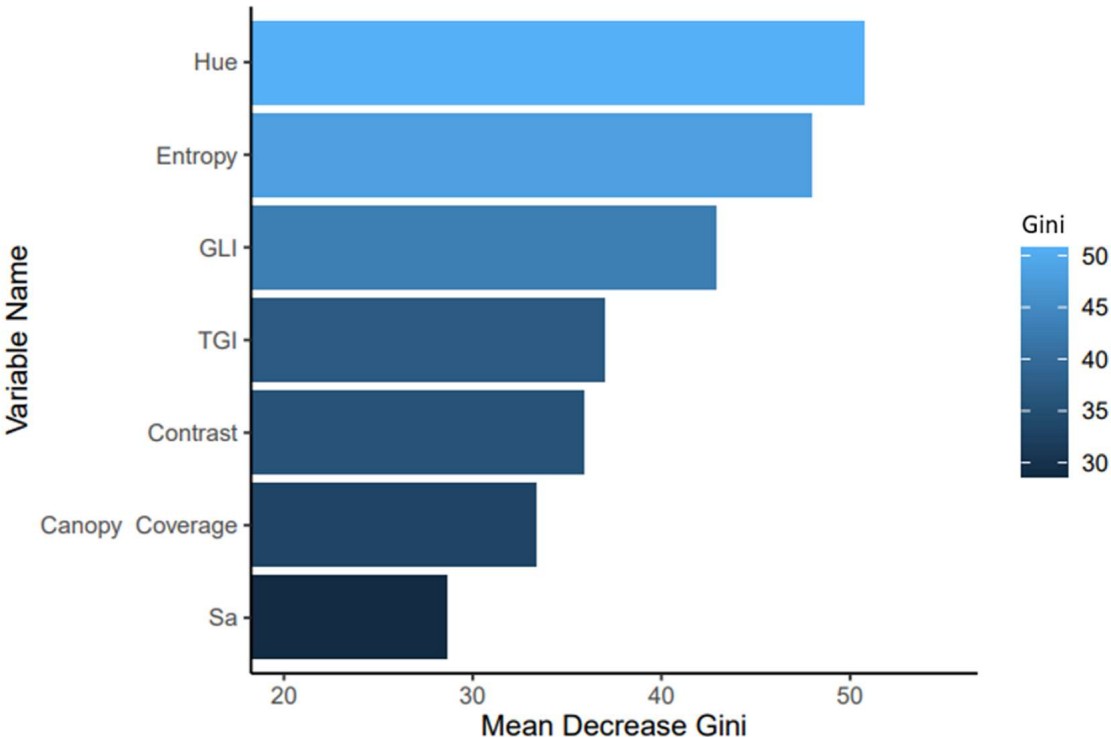

**Figure 4.** Feature importance represented as the mean decrease Gini coefficient for the seven image features included in the RF model.

## 4. Discussion and Conclusions

Unmanned aerial vehicles (UAVs) have been used in agricultural research to collect high-resolution and spectral images from field plots, of which multiple image features and vegetation indices can be extracted and later used to explain observed phenotypic variation. With a hyperspectral plant-sensing hand device, Huang et al. (2016) reported differentiation between treated (dicamba application) and non-treated soybean with accuracies ranging from 76 to 86%, demonstrating the potential of remote sensing in detecting early dicamba injury in soybean [33]. Out of the seven extracted vegetation indices, anthocyanin reflectance index (ARI) [57] and photochemical reflectance index (PRI) [58] were the most significant to differentiate between treated and non-treated soybean (Huang et al., 2016). Zhang et al. (2019) combined 21 sensitive spectral features with three machine learning algorithms (naive Bayes (NB), RF, and support vector machine (SVM)) to assess soybean damage from dicamba. Reported overall accuracies ranged from 0.69 to 0.75, with RF algorithm having the highest overall accuracy (0.75) [35]. These reports are compatible with our findings, particularly the highest accuracy obtained from RF algorithms. The superiority of RF may be associated with its ability to handle high data dimensionality and multicollinearity [35,59]. In addition, RF often performs better than ANN with limited training sets [60]. Abrantes et al. (2021) explored six RGB-derived image features to correlate with visual injury ratings and found the modified green–red vegetation index (MGRVI, $R^2 = 0.93$) [61], modified photochemical reflectance index (MPRI, $R^2 = 0.93$) [58], and excess green (ExG, $R^2 = 0.89$) [62] to be the indices with the highest correlation with dicamba damage [36]. Marques et al. (2021) found the triangular greenness index to be highly correlated with dicamba dosage, with $R^2$ ranging from 0.71 to 0.94 based on days after application [37].

In our study, seven image features were used to classify the visual assessment of dicamba injuries reported as categorical variables (tolerant, moderate, and susceptible). ANOVA results showed significant differences among dicamba classes, although no significant effect was found across locations, indicating a rather homogeneous and uniform off-target dicamba distribution in the testing area. Considering that each field represented a unique and diverse environment with variable soil types, planting dates, and soybean genotypes, these results are promising when it comes to the relevance of these image features to represent injuries caused by off-target dicamba as well as applying and replicating the developed classification machinery in non-tested soybean genotypes and environments. Interestingly, the observed variation for each image feature was consistent with physiological consequences of dicamba damage, as well as expected variations based on overall symptoms of the injuries. For instance, based on our field observations, the severe cupping of the immature leaves drastically reduced vegetative growth and canopy coverage. The image feature representing standardized canopy coverage clearly distinguished the three dicamba response classes, of which the tolerant group had the highest mean value (0.616, a) followed by moderate (0.191, b) and susceptible (−0.709, c) classes. The vegetation index GLI, which represents the photosynthetically functional component of the leaf area index [63], had the highest mean value for the tolerant class (0.455, a) and lowest for the susceptible class (−0.798, c). In addition to canopy coverage and overall vegetation development, color and texture-based image features also significantly differentiated the dicamba response classes. The cupping of the immature leaves produced a low-intensity, lighter green color as well as a rough texture to the plants. Entropy, a feature used in this study to represent leaf texture, describes how much information is provided by an image [64–66]. From the physical measurement point of view, higher entropy indicates more information being transmitted and therefore a smoother, higher quality image [67]. The tolerant class showed the highest mean value for entropy (0.512, a), followed by moderate (0.267, b) and susceptible (−0.856, c) classes. Hue and saturation, which indicate the color and intensity of an image [68], significantly distinguished between tolerant and susceptible classes, of which the tolerant class had higher mean values than the susceptible (0.211 and 0.295 vs −0.654 and −0.222, respectively). This means that, as expected, tolerant lines presented a more

intense or darker green color as opposed to low-intensity, lighter green color observed in susceptible plants. It is also important to point out that our studies were conducted in five 6-acre fields on a 700+ acre farm with prolonged off-target dicamba exposure, compared to previous studies where direct spray treatment was compared with non-treated controls. In addition, we used 230 diverse soybean lines with different genetic backgrounds in a naturally volatile environment; the phenotypic differences observed should reflect underlining genetic differences of testing lines in response to dicamba, whereas previous studies focused on phenotypic differences caused by the herbicide application treatments rather than genotypes. Moreover, multiple image features were used in our study and showed consistent results in differentiating the phenotypic responses and classifying lines into distinctive classes. Overall, our results show that the technology offers plant breeders a rapid, simple, precise, and efficient tool in breeding and selection for natural tolerance to dicamba.

The development and commercialization of genetically modified DT soybean were followed by widespread reports of off-target damage in non-DT soybean and multiple dicots plant species [12–16]. As a growth regulator herbicide, dicamba is a synthetic auxin that triggers abnormal growth and/or plant death in sensitive dicots [69]. Besides the growth stage, dosage, frequency, and duration of the exposure, the genetic background of soybean genotypes may potentially affect the observed symptoms. This research represents the first large-scale screening of genetically diverse soybean to off-target dicamba damage, including 230 elite breeding lines derived from 115 unique biparental populations. The developed classification machinery precisely distinguished tolerant and susceptible soybean genotypes with diverse genetic backgrounds. In addition, image features and overall classification accuracies showed consistency across several unique environments, which reinforces the ability of this classification machinery to accurately classify non-tested genotypes in non-tested environments. Further studies are encouraged, using multi and hyperspectral images, as well as controlled dicamba dosages and exposure durations, to investigate potential enhancements in classification accuracies. As a cost-effective choice, this UAV RGB-based imagery system can be implemented in plant breeding programs targeting the identification and selection of genotypes showing enhanced tolerance to off-target dicamba damage.

**Author Contributions:** Conceptualization, C.C.V., S.S., F.T., J.Z. (Jing Zhou), J.Z. (Jianfeng Zhou) and P.C.; Data curation, C.C.V., S.S. and F.T.; Formal analysis, C.C.V., S.S., F.T. and J.Z. (Jing Zhou); Funding acquisition, P.C.; Investigation, C.C.V., S.S., F.T., J.Z. (Jing Zhou), D.J., H.T.N., J.Z. (Jianfeng Zhou) and P.C.; Methodology, C.C.V., S.S., F.T., J.Z. (Jing Zhou), J.Z. (Jianfeng Zhou) and P.C.; Project administration, P.C.; Resources, J.Z. (Jianfeng Zhou) and P.C.; Software, S.S. and F.T.; Supervision, J.Z. (Jianfeng Zhou) and P.C.; Validation, D.J., H.T.N., J.Z. (Jianfeng Zhou) and P.C.; Visualization, C.C.V., S.S., F.T., J.Z. (Jing Zhou), J.Z. (Jianfeng Zhou) and P.C.; Writing—original draft, C.C.V., S.S., F.T. and J.Z. (Jing Zhou); Writing—review & editing, C.C.V., S.S., F.T., J.Z. (Jing Zhou), D.J., H.T.N., J.Z. (Jianfeng Zhou) and P.C. All authors have read and agreed to the published version of the manuscript.

**Funding:** This research was funded by Mid-south Soybean Board (20-455-21), United Soybean Board (2120-172-0147), and Missouri Soybean Merchandising Council (20-455-21).

**Data Availability Statement:** The data presented in this study are available on request from the corresponding author.

**Acknowledgments:** The authors would like to thank the University of Missouri—Fisher Delta Research, Extension, and Education Center soybean breeding team for their technical support in preparing and conducting the field trials. The authors would also like to extend their gratitude to the funding agencies that supported this work including the Mid-south Soybean Board, United Soybean Board, and Missouri Soybean Merchandising Council.

**Conflicts of Interest:** The authors declare that the research was conducted in the absence of any commercial or financial relationships that could be construed as a potential conflict of interest.

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
