# Peer review of "Differentiate Soybean Response to Off-Target Dicamba Damage Based on UAV Imagery and Machine Learning"

_remotesensing, doi:10.3390/rs14071618_

Round 1

Reviewer 1 Report

This study aims to develop a method to differentiate soybean response to dicamba using unmanned aerial vehicle-based imagery and machine learning models. The proposal provides a very interesting vision, as this method would be a very useful resource for engineers. The manuscript is well-structured with novelty. I suggest acceptance after minor edits.

  • Image characteristics are affected by the change of environmental light, how is the situation handled?
  • It is recommended to add more pictures taken by the UAV and pictures of three feature categories.
  • Vision technology applications in various engineering fields should also be introduced for a full glance of the scope of related areas. The first paragraph introducing the research topic may present a much broad and comprehensive view of the problems related to your topic with citations to authority references:
    Wu, F.; Duan, J.; Chen, S.; Ye, Y.; Ai, P.; Yang, Z. Multi-Target Recognition of Bananas and Automatic Positioning for the Inflorescence Axis Cutting Point. Frontiers in Plant Science 2021, 12:705021. 
    Tang, Y.; Zhu, M.; Chen, Z.; Wu, C.; Chen, B.; Li, C.; Li, L. Seismic Performance Evaluation of Recycled aggregate Concrete-filled Steel tubular Columns with field strain detected via a novel mark-free vision method. Structures, 2022, 37: 426-441.                                            Chen, Z.; Wu, R.; Lin, Y.; Li, C.; Chen, S.; Yuan, Z.; Chen, S.; Zou, X. Plant Disease Recognition Model Based on Improved YOLOv5. Agronomy 2022, 12, 365.

Author Response

  • Image characteristics are affected by the change of environmental light, how is the situation handled?

R: Thank you very much for the question regarding the environmental light issue in obtaining UAV images and we agree with the issue as well. Images and visual scoring of different fields were acquired on the same day or one day before or on the next day of imaging. As several days of images were included in the dataset obtained from five different fields, it was expected that there would be environmental light differences. By considering this issue, we standardized the image features before analyzing them with ANN and RF. Standardizing image features data are computationally effective and includes tolerable statistical errors resulting in a more efficient and robust classification of phenotyping traits (Singh and Singh, 2020). Detailed information regarding the standardization was included in the manuscript: Lines 175-184.

References: Singh, D. & Singh, B. (2020). Investigating the impact of data normalization on classification performance. Applied soft computing. 97(Part B), 105524.

  • It is recommended to add more pictures taken by the UAV and pictures of three feature categories.

R: Thank you for pointing this out. Although we completely agree that visual demonstration of three categories is essential, we thought that the current figure does represent the overall phenotypic differences among classes well, and adding additional examples would not necessarily improve the readability of that section of the manuscript. Please let us know if this is reasonable, otherwise, we are happy to share additional information.

  • Vision technology applications in various engineering fields should also be introduced for a full glance of the scope of related areas. The first paragraph introducing the research topic may present a much broad and comprehensive view of the problems related to your topic with citations to authority references:
    Wu, F.; Duan, J.; Chen, S.; Ye, Y.; Ai, P.; Yang, Z. Multi-Target Recognition of Bananas and Automatic Positioning for the Inflorescence Axis Cutting Point. Frontiers in Plant Science 2021, 12:705021. 
    Tang, Y.; Zhu, M.; Chen, Z.; Wu, C.; Chen, B.; Li, C.; Li, L. Seismic Performance Evaluation of Recycled aggregate Concrete-filled Steel tubular Columns with field strain detected via a novel mark-free vision method. Structures, 2022, 37: 426-441.
    Chen, Z.; Wu, R.; Lin, Y.; Li, C.; Chen, S.; Yuan, Z.; Chen, S.; Zou, X. Plant Disease Recognition Model Based on Improved YOLOv5. Agronomy 2022, 12, 365.

R: Thank you for the suggestions. The recommended publications all have merit in their own field, but we believe these do not necessarily align with the goal and objective of our research. We aimed to keep the introduction as straightforward as possible including information directly related to the methodology, crop, and overall target problem of this research.

Reviewer 2 Report

Dear colleagues,
Thank you very much for the interesting paper. You have highlighted the importance of the topic very well. One point that may be a bit difficult for many readers to grasp is the differences in the genotypes they studied. These differences end up leading to the misclassifications. The remote sensing approach based on the drone images is quite simplistic. On the other hand, I can definitely understand that they rely on a "simple" RGB drone. After all, this is inexpensive and also provides very good images. However, they do without a highly accurate georeferencing. As a result, they also miss out on an important parameter, the growth height, which can be extracted from the drone images. This parameter, moreover, strongly correlates with biomass. 
However, it is not entirely clear why they chose exactly these 7 parameters. There are many more, most of which can be calculated very easily. By the way, the number of parameters does not represent a big additional effort in the context of a Random Forest classification. During RF classification, it is necessary to calculate the Gini index in order to identify the most meaningful parameters, or to eliminate those that contribute little or nothing to the success of the classification. Please report on that, even if you stick to your seven parameters.

Author Response

  • Inclusion of height as a model feature.

R: Dear reviewer, thank you for the positive response and very constructive feedback. As much as plant height can be correlated with biomass within a growth habit, it would not necessarily benefit our study. In this research, we have soybean genotypes from three different growth habits namely determinate, semi-determinate, and indeterminate. Consequently, these differ drastically in plant height but this difference does not translate into more biomass and/or better response to dicamba. Therefore, canopy size was included as a direct measurement of crop productivity (biomass) and response to dicamba that is independent of plant height (and therefore of growth habit).

  • Why only seven image features have been used?

R: Image features were selected based on the visual observed symptoms of dicamba as well as previous experience in computing and predicting different soybean traits using UAV- imagery technique and machine learning methods. For instance, dicamba damage affects the texture (herein represented as entropy and contrast), color intensity (represented as hue and saturation), canopy size, and overall crop health and development (represented as TGI and GLI). In addition, we completely agree with the reviewer that more features could be easily calculated and included in the analysis. However, too much of variables or features could cause multicollinearity (Lieberman & Morris, 2014) as the resulting data sets may contain many phenotypic traits or redundant features which are correlated with each other (Chen et al. 2014). To avoid this possible situation only seven image-derived features were selected and analyzed to classify the dicamba tolerance effect on soybean. Details of each feature and its rationale can be found in lines 241-258 and 341-363.

References:

Lieberman, M. G. and Morris, J. D. (2014). The precise effect of multicollinearity on classification prediction. Multiple Linear Regression Viewpoints, 40(1), 5-10.

Dijun Chen, Kerstin Neumann, Swetlana Friedel, Benjamin Kilian, Ming Chen, Thomas Altmann, Christian Klukas, Dissecting the Phenotypic Components of Crop Plant Growth and Drought Responses Based on High-Throughput Image Analysis, The Plant Cell, Volume 26, Issue 12, December 2014, Pages 4636–4655, https://doi.org/10.1105/tpc.114.129601

  • During RF classification, it is necessary to calculate the Gini index in order to identify the most meaningful parameters

The figure 4 (attached document) shows the mean decrease Gini coefficients of the seven image features in the random forest classification model. The coefficients indicate how much accuracy the model losses by excluding each variable and the higher value of mean decrease Gini, the higher the importance of the variable in the model (Han et al. 2016). Detailed information regarding these results was added to the manuscript in lines 312-324.

References:

Hong Han, Xiaoling Guo and Hua Yu, "Variable selection using Mean Decrease Accuracy and Mean Decrease Gini based on Random Forest," 2016 7th IEEE International Conference on Software Engineering and Service Science (ICSESS), 2016, pp. 219-224, doi: 10.1109/ICSESS.2016.7883053.

Reviewer 3 Report

Remote sensing technologies are expected to revolutionize agriculture and especially Unmanned Aerial Vehicles (UAVs) have been proven very effective in a variety of applications related to crops management, by capturing high spatial and temporal resolution images.

This work focuses on soybean production, and more precisely on differentiating soybean response to off-target dicamba damage based on UAV Imagery. Machine Learning algorithms are exploited for this purpose, with the paper utilizing and comparing two of the most widely used systems, i.e. artificial neural network (ANN) and random forest (RF) algorithms

As mentioned in the paper, the assessment of injuries caused by off-target dicamba exposure is usually done by differentiating the crops in several categories representing the level of injuries or directly assessing the percentage of injury (0 - 100%). In the current work, the authors use categorical variables and exploit classification and not regression methods.

This article is in a very important and developing research field. Especially the fact that they use low-cost systems (e.g. RGB camera and not expensive hyperspectral ones), having promising results. 

The results show that the specific approach is promising and has the potential to be utilized in real applications.  However, the results are not significantly different and improved compared to other results presented in the literature.

In addition, there are some flaws in the presentation of the experimental evaluation.

First of all, the authors refer to R2 results for their classification. In general, R2 is not a good measure to assess classification results. R2 is suitable for predicting continuous variables and is one of the important evaluation measures for regression-based machine learning models. Based on the context, I assume that with the mentions in R2, they falsely refer to the overall classification accuracy of their systems.

Moreover, they measure the Class Accuracy, for which they use the whole set of instances, by taking into account both the true positive and true negative (and false positive and false negative) instances. Usually, classification accuracy is calculated for the entire dataset and not for individual classes. If someone wants a measure for the accuracy in individual classes, the metric of precision can be used, as it is calculated in a similar way.

Taking into account the instances of the entire dataset, classes with a higher amount of instances than others will bias the results for the individual class accuracies. This happens in the presented results with the second class, increasing the individual class accuracy for all classes, although the actual precision is sometimes quite low.

In addition, I would like to see some justifications from the authors for their choices, like the architecture of their ANNs, or the features they used for their dataset. As they mention in the paper, there have been proposed in recent works some RGB-derived image features to correlate with visual injury ratings, with very promising results, like MGRVI, MPRI, and ExG. Why do the authors prefer to not use these vegetation indices?

To summarise, I think that the work is promising, as it addresses an important problem, being in a research area that has attracted a lot of attention recently, with many real applications. However, in order to publish this work, I believe that major revisions should be made, addressing the issues I mentioned above.

Author Response

Dear Reviewer, we deeply appreciate your thoughtful and constructive feedback. We believe all raised concerns have been addressed and the manuscript is improved. Detailed responses are below - we also attached the complete revision responses for all reviewers. Please let us know if any additional clarification is needed. Once again, thank you for helping us improve our manuscript.

  • First of all, the authors refer to R2 results for their classification. In general, R2 is not a good measure to assess classification results. R2 is suitable for predicting continuous variables and is one of the important evaluation measures for regression-based machine learning models. Based on the context, I assume that with the mentions in R2, they falsely refer to the overall classification accuracy of their systems.

R: Dear reviewer, we appreciate your positive comments and constructive feedback. Thank you for pointing out the misuse of R2 in the abstract, this has been edited accordingly in line 25.

  • Moreover, they measure the Class Accuracy, for which they use the whole set of instances, by taking into account both the true positive and true negative (and false positive and false negative) instances. Usually, classification accuracy is calculated for the entire dataset and not for individual classes. If someone wants a measure for the accuracy in individual classes, the metric of precision can be used, as it is calculated in a similar way. Taking into account the instances of the entire dataset, classes with a higher amount of instances than others will bias the results for the individual class accuracies. This happens in the presented results with the second class, increasing the individual class accuracy for all classes, although the actual precision is sometimes quite low.

R: Thank you for pointing this out. We completely agree that classification accuracy alone can be highly misleading, especially when the data is unbalanced. This is why we have considered four additional metrics and the confusion matrix which can be seen in Tables 3 and 4, and more detailed on lines 212-215. We have expanded the description of the results including more details over the precision and other metrics as seen in lines 280-283 and 302-307. It is also interesting to observe that, although precision may be quite low for the tolerant class, the misclassification is often between tolerant and moderate, and rarely a tolerant line will be classified as susceptible and vice-versa.

  • In addition, I would like to see some justifications from the authors for their choices, like the architecture of their ANNs, or the features they used for their dataset. As they mention in the paper, there have been proposed in recent works some RGB-derived image features to correlate with visual injury ratings, with very promising results, like MGRVI, MPRI, and ExG. Why do the authors prefer to not use these vegetation indices?

R: Thank you for pointing this out. As also pointed out by reviewer 2, the rationale behind the selected image features were based on the overall visual symptoms of off-target dicamba damage. For instance, dicamba damage affects the texture (herein represented as entropy and contrast), color intensity (represented as hue and saturation), canopy size, and overall crop health and development (represented as TGI and GLI). Details of each feature and its rationale can be found in lines 245-262 and 360-382.

To summarise, I think that the work is promising, as it addresses an important problem, being in a research area that has attracted a lot of attention recently, with many real applications. However, in order to publish this work, I believe that major revisions should be made, addressing the issues I mentioned above.

R: Once again, we appreciate your suggestions and comments and believe most of the raised concerns have been addressed. We thought it was important to reinforce that, although the results may not be significantly different from other reports in the literature, the overall methodology and application of this work do differ substantially from what is available in the literature. Starting from the plant materials, we used a vast number of genotypes covering a wide range of genetic diversity. This not only enhances the statistical power of this work but significantly expands the application of these results across multiple genotypes in future studies and/or usability in breeding programs. In addition, the exposure of dicamba being completely natural and prolonged represents a very unique scenario to reproduce what is happening in the “real world”. All reports to this date are based on extremely limited genetic diversity and pre-determined dosages of dicamba that do not necessarily represent what farmers/researchers would get exposed to.

Round 2

Reviewer 2 Report

Dear Colleagues,

thank you for your detailed reply and the additional information in the paper.

Reviewer 3 Report

Dear authors,

thank you for the detailed reply.

I find that the article is improved from the last version. Although there are not too many changes in the paper, the ones made, improve the presentation in some important parts.

Some thoughts on how it could be improved more:

I still didn't see why the specific variables were chosen to train the model, instead of others. I see the relevance of the variables to the desired classification, but I didn't understand why others were excluded, although it would be quite easy to calculate them (i.e. the vegetation indices I mentioned in the previous review). Maybe the Gini index for these variables, compared to the ones chosen, would help in the presentation.

I would also like to see some informative parts of the reply (not only on my comments) included in the paper, as they were quite helpful.

Although I think that these minor changes could improve the paper, I don't think they are significant enough to have another round of review just for them, they are just suggestions. I believe that the article can be accepted now in its current form